# Shotgun metagenomic analysis of saliva microbiome suggests *Mogibacterium* as a factor associated with chronic bacterial osteomyelitis

**Hiroko Yahara** [1]*, **Souichi Yanamoto** [2], **Miho Takahashi** [3], **Yuji Hamada** [3], **Takuya Asaka** [4], **Yoshimasa Kitagawa** [4], **Kuniyasu Moridera** [5], **Kazuma Noguchi** [5], **Yutaka Maruoka** [6], **Koji Yahara** [7]*

1 Genome Medical Science Project, Research Institute, National Center for Global Health and Medicine, Tokyo, Japan, 2 Department of Oral Oncology, Graduate School of Biomedical and Health Sciences, Hiroshima University, Hiroshima, Japan, 3 Department of Oral and Maxillofacial Surgery, Tokai University Hachioji Hospital, Tokyo, Japan, 4 Department of Oral Diagnosis and Medicine, Hokkaido University Graduate School of Dental Medicine, Sapporo, Japan, 5 Department of Oral and Maxillofacial Surgery, School of Medicine, Hyogo Medical University, Hyogo, Japan, 6 Department of Oral and Maxillofacial Surgery, Center Hospital of the National Center for Global Health and Medicine, Tokyo, Japan, 7 Antimicrobial Resistance Research Center, National Institute of Infectious Diseases, Tokyo, Japan

* h-yahara@ri.ncgm.go.jp (HY); k-yahara@niid.go.jp (KY)

## Abstract

Osteomyelitis of the jaw is a severe inflammatory disorder that affects bones, and it is categorized into two main types: chronic bacterial and nonbacterial osteomyelitis. Although previous studies have investigated the association between these diseases and the oral microbiome, the specific taxa associated with each disease remain unknown. In this study, we conducted shotgun metagenome sequencing ($\geq$10 Gb from $\geq$66,395,670 reads per sample) of bulk DNA extracted from saliva obtained from patients with chronic bacterial osteomyelitis (N = 5) and chronic nonbacterial osteomyelitis (N = 10). We then compared the taxonomic composition of the metagenome in terms of both taxonomic and sequence abundances with that of healthy controls (N = 5). Taxonomic profiling revealed a statistically significant increase in both the taxonomic and sequence abundance of *Mogibacterium* in cases of chronic bacterial osteomyelitis; however, such enrichment was not observed in chronic nonbacterial osteomyelitis. We also compared a previously reported core saliva microbiome (59 genera) with our data and found that out of the 74 genera detected in this study, 47 (including *Mogibacterium*) were not included in the previous meta-analysis. Additionally, we analyzed a core-genome tree of *Mogibacterium* from chronic bacterial osteomyelitis and healthy control samples along with a reference complete genome and found that *Mogibacterium* from both groups was indistinguishable at the core-genome and pangenome levels. Although limited by the small sample size, our study provides novel evidence of a significant increase in *Mogibacterium* abundance in the chronic bacterial osteomyelitis group. Moreover, our study presents a comparative analysis of the taxonomic and sequence abundances of all genera detected using deep salivary shotgun metagenome data. The distinct enrichment of *Mogibacterium* suggests its potential as a marker to

**Data Availability Statement:** All relevant data are within the manuscript and its Supporting information files.

**Funding:** This study was supported by a Grant-in-Aid for Scientific Research of Education, Culture, Science, Sports, and Technology (MEXT) from Japan (19J40070 to H.Y.). This work was supported in part by Grants-in-Aid for Research from the National Center for Global Health and Medicine (23A3001). he funders had no role in study design, data collection and analysis, decision to publish, or preparation of the manuscript.

**Competing interests:** The authors declare that they have no competing interests.

distinguish between patients with chronic nonbacterial osteomyelitis and chronic bacterial osteomyelitis, particularly at the early stages when differences are unclear.

## Introduction

Osteomyelitis of the jaw is a severe inflammatory disorder that affects the bones, and it is challenging to treat owing to the high recurrence rate of the chronic forms. However, the lack of international consensus on the definitions of these forms hampers their diagnosis. Similarly, differences in the treatment strategies reflect an inadequate understanding of the predisposing factors and processes leading to jaw osteomyelitis.

Osteomyelitis of the jaw is not a unique condition, and two main types have been described in the literature. The first type is characterized by an apparent odontogenic infectious etiology and is typically defined as secondary chronic osteomyelitis [1]. Herein, we refer to it as chronic bacterial osteomyelitis, where identifiable infectious pathogens are present, leading to suppurative variants characterized by the presence of pus, abscesses, fistulas, and/or sequestrations [2]. The second type of osteomyelitis is the non-suppurative chronic variant, which is defined as a chronic inflammatory disorder of unknown etiology [3]. Herein, we refer to this condition as chronic nonbacterial osteomyelitis (CNO), which encompasses primary chronic osteomyelitis, diffuse sclerosing osteomyelitis, juvenile mandibular chronic osteomyelitis, and chronic recurrent multifocal osteomyelitis. Although CNO does not exclude the presence of pathogens, it is considered a chronic variant without suppurative characteristics [4].

The influence of genetic and immunological backgrounds on CNO is not completely understood. A recent study defined the alleles of all 35 human leukocyte antigen (HLA) loci and haplotype structures of the killer cell immunoglobulin-like receptor (KIR) region and identified a specific amino acid substitution in HLA-C in combination with the telomeric KIR genotype, which had a significantly higher frequency in the CNO population compared to the control population [5]. Another recent study investigated CNO based on whole blood RNA sequencing (>6 Gb per sample) of 11 patients and 9 healthy controls in Japan and employed a recently developed method suitable for small datasets. This study revealed subnetworks of genes underlying patient characteristics and identified the gene encoding glycophorin C with the highest discrimination ability [6].

Although CNO is an inflammatory disorder, the mechanisms triggering this inflammatory process are poorly understood. Several authors have reported negative bacterial culturing from biopsies, indicating a nonbacterial origin [3, 7]. The inflammatory nonsuppurative process supports the suspicion of an autoimmune etiology. Recent studies have highlighted the role of the microbiome in the pathogenesis of autoimmune diseases [8, 9]. These studies suggest that the microbiome not only directly influences the pathogenesis of osteomyelitis but also indirectly affects it through biochemical signals produced by the microbiota of non-osseous tissues that trigger cells and microbes within the osteomyelitis tissue. A recent pilot study focusing on chronic recurrent multifocal osteomyelitis, a severe systemic type of CNO involving multifocal autoinflammatory bone lesions, applied 16S rRNA gene sequencing to oral swab samples from patients. This study reported a shift in the composition of the oral microbiome among patients treated with different medications, although specific taxa significantly associated with the disease were not identified or discussed [10].

Several studies have previously explored the associations between chronic bacterial osteomyelitis and microbial infection, as summarized in a recent review [11]. For example, an *Escherichia coli* strain exhibiting multiple antibiotic resistance was cultured from bilateral

maxillary osteomyelitis of a patient with diabetes [12]. *Actinomyces* identified from patients with maxillary osteomyelitis were suggested to be derived from pulpal or periodontal infections [13]. Another study that sampled infected bone detected three predominant commensal anaerobic strains (*Parvimonas micra*, *Staphylococcus* spp., and *Fusobacterium nucleatum*) through PCR analysis [14]. Additionally, 16S rRNA gene sequencing of bone samples of different types of osteomyelitis, including chronic bacterial osteomyelitis (suppurative osteomyelitis) and CNO, reported that the core microbiome is predominantly composed of anaerobic microbes, such as *Fusobacterium nucleatum*, *Tannerella* sp., and *Porphyromonas* sp. [15].

However, approaches based on bacterial culture and PCR have limitations and may miss unculturable microbes associated with the disease. Although 16S rRNA gene sequencing is widely used, it also has limitations, such as the introduction of biases from several sources, including PCR amplification, primer design, and sequencing artifacts [16]. Furthermore, the previous study that employed 16S rRNA gene sequencing on bone samples [15] did not distinguish between chronic bacterial osteomyelitis and CNO and instead focused on reclassification into three clinical stages. A recent review citing a previous 16S rRNA study concluded that no conclusive microbiome analyses of chronic bacterial osteomyelitis of the jaw have been reported [11]. Therefore, the contribution of the increased prevalence of specific microbial taxa in the oral microbiota to osteomyelitis remains unknown. Additionally, 16S rRNA gene sequencing does not provide information on the presence, abundance, or function of specific genes in the microbiome.

To address these current challenges, we conducted shotgun metagenome sequencing ($\geq$10 Gb per sample) of bulk DNA extracted from the saliva of patients with chronic bacterial osteomyelitis and CNO. We then compared the metagenome data with those of healthy controls in Japan from our previous study [17], which were generated using the same sample collection and DNA extraction protocols as the present study and are publicly available. Through comparative quantitative analyses of the taxonomic compositions of these deep metagenomes in terms of both taxonomic and sequence abundances, we found a statistically significant increase in the abundance of *Mogibacterium* in chronic bacterial osteomyelitis, which suggests its potential as a marker to distinguish between patients with CNO and chronic bacterial osteomyelitis.

## Methods

### Diagnostic criteria

Although various diagnostic techniques are employed to assess osteomyelitis, the consensus among authors is that the final diagnosis should be based on multiple criteria, including clinical presentation, patient history, and imaging techniques [18]. In cases of chronic osteomyelitis, the presence of infectious pathogens can be observed in different stages, namely in pus, abscess/fistula, and sequestration. In most of these cases, there is an apparent odontogenic or infectious etiology, typically resulting from trauma that introduces pathogens into the tissues. In this study, this condition was defined as chronic bacterial osteomyelitis.

The diagnostic criteria for CNO were consistent with those used in our previous study [5]: 1) recurrent pain and swelling; 2) radiographic appearance of a mixed pattern of sclerosis and osteolysis, and uptake of scintigraphic agents (such as technetium 99 m) in the jawbone; 3) limited or no response to antibiotic treatment; and 4) increased bone resorption and deposition, along with varying degrees of bone sclerosis and medullary fibrosis, without suspicion of malignancy.

## Sample collection, DNA extraction, and metagenome sequencing

In total, 15 Japanese patients were prospectively enrolled in this study, including 5 with chronic bacterial osteomyelitis and 10 with CNO (S1 Table). Based on previous reports that microbial profiles within subjects were stable throughout a 24-h period [19] and that similar profiles could be obtained from unstimulated and stimulated saliva [20], 1 mL of unstimulated saliva was collected and stored using a specialized kit for microbial and viral DNA/RNA (OMNIgene ORAL OM-501), following established protocols [17, 21]. The collected samples were used for DNA extraction and metagenomic sequencing.

DNA extraction from the saliva samples was performed using an enzymatic method that was previously applied to saliva samples [17, 22]. The extracted DNA samples were stored in 50 μL of pure water and used for library construction and genome sequencing using the Illumina HiSeq $2 \times 150$ bp paired-end run protocol. The raw sequence data generated for the patients ranged from 10.0 to 22.8 Gb (S1 Table). The sequencing depth per base averaged 74. Metagenomic data of five healthy controls that were previously sequenced and are publicly available [17] (S1 Table) were included in this study.

## Preprocessing, taxonomic profiling, and functional profiling

The EDGE pipeline version 1.5 [23] was used for preprocessing (trimming or filtering out reads and removing reads mapped to the human genome) of the HiSeq data. The initial quality control step discarded 0.06–1.54% of reads and trimmed 0.06–1.54% of bases. Subsequently, 0.09–31.1% of the filtered reads that mapped to the human genome were removed. Subsequently, the lowest number of remaining read pairs among the samples was 27,478,389 (8.2 Gb). To account for differences in sequence coverage, we randomly selected the same number of read pairs from each sample [24] for subsequent analyses using BBmap, which is included in the BBTools software package [25].

Taxonomic profiling was conducted using MetaPhlAn3 [26] and Kraken2 [27] followed by Bracken [28] to estimate the "taxonomic abundance" (calculated as the number of genomes (single-copy marker genes) of a given taxon relative to the total number of genomes detected) and "sequence abundance" (calculated as the proportion of sequence reads assigned to a given taxon out of the total number of sequence reads) [29]. As MetaPhlAn3 treats paired reads independently, we concatenated the read pairs into a single input file in advance using BBmap while subsampling the reads [25]. Sequence abundance refers to the fraction of sequence reads assigned to each taxon in the reference genome database and depends on the genome size of the taxon, potentially resulting in under-or overestimation. In contrast, taxonomic abundance is the ratio of the sequence coverage of single-copy marker genes of each taxon to that of all taxa. This approach avoids underestimation or overestimation caused by variations in genome size [29]. In this study, we primarily used the taxonomic abundance as our main metric. Functional profiling was conducted using HUMAnN 3.0 [26].

## Assembly, taxonomic analysis of contigs, and gene-by-gene analysis

HiSeq reads were assembled using SPAdes [30] with the "—meta" option. The Contig Annotation Tool (CAT) [31] was employed for taxonomic classification of the contigs based on amino acid sequence searching of each open reading frame (ORF) against the NCBI nr database. The classification was determined by summing all scores, supporting a certain taxonomic classification (superkingdom, phylum, class, order, family, genus, and species) from ORFs separately, and determining whether the summation exceeded a cutoff value (by default, 0.5).

Because we focus on *Mogibacterium* as a characteristic genus associated with chronic bacterial osteomyelitis in the Results section, nucleotide sequences of contigs assigned to

*Mogibacterium* were selected and protein-coding genes were predicted for each contig using Prokka [32]. A reference complete genome sequence of *Mogibacterium diversum* strain CCUG 47132 [33] (NCBI assembly accession GCF_002998925.1) was downloaded, and Prokka was employed to predict the protein-coding genes. We then constructed a gene presence or absence matrix for the entire set of genes (i.e., orthologous clusters) detected among the reference complete genome, chronic bacterial osteomyelitis, and healthy control samples and a core-genome alignment using the Roary pipeline [34] with the "-i 90—group_limit 1000000" option. Subsequently, we constructed a maximum likelihood tree from this alignment using PhyML [35]. This tree, along with metadata, was illustrated using Phandango [36].

## Statistical analysis

Associations between taxonomic or functional profiles and host groups (healthy controls, CNO, and chronic bacterial osteomyelitis) were statistically tested using MaAsLin2 [37], which accounts for zero-inflated, high-dimensional, and extremely non-normal microbiome data. The healthy control group was specified as a reference, and the difference between the CNO and healthy control groups and that between the chronic bacterial osteomyelitis and healthy control groups were tested. According to previous studies [38, 39], we specified the total sum scaling (TSS) normalization (to ensure that the profile values ranged from 0 to 1) and arcsine square-root transformation (AST) (to stabilize the variance and improve parametric estimation models in the presence of violated data assumptions, such as normality and homoscedasticity) [37]. Box plots were created using JMP Pro 14 [40]. All other statistical analyses were conducted using R software (version 4.2.2) [41], and the Venn diagram was created using the "eulerr" package.

## Ethics approval and consent to participate

This study was approved by the ethics committees of the Research Institute National Center for Global Health and Medicine (approval number NCGM-A-003228), Nagasaki University (20191101), Tokai University (19R-075), Hokkaido University (2019–1), Hyogo College of Medicine (0419), and National Institute of Infectious Diseases (1283). The study design followed the Declaration of Helsinki guidelines, and written informed consent was obtained from all the participants. The recruitment period started on May 10, 2019, following approval from the ethics committee of the Research Institute National Center for Global Health and Medicine (the main facility of this project), and ended on March 31, 2022.

## Results

### Taxonomic profiling reveals potential microbes associated with chronic bacterial osteomyelitis

Among the 74 bacterial genera detected in at least one sample (S2 Table), only four exhibited statistically significant differences in taxonomic abundance ($P_{FDR}$<0.05, tested using a model with normalization and transformation implemented in MaAsLin2) among the CNO, chronic bacterial osteomyelitis, and healthy control groups (Fig 1). *Abiotrophia*, *Lautropia*, and *Granulicatella* showed a significant decrease in the CNO and chronic bacterial osteomyelitis groups (Fig 1a), whereas *Mogibacterium* showed a significant increase in the chronic bacterial osteomyelitis group (Fig 1b). *Abiotrophia*, *Lautropia*, and *Granulicatella* were detected in all healthy control samples, and their respective taxonomic abundances were consistently below 3%. In contrast, these genera were not detected in 40–60% of the CNO samples and 40–80% of the chronic bacterial osteomyelitis samples. Meanwhile, the average taxonomic abundances of

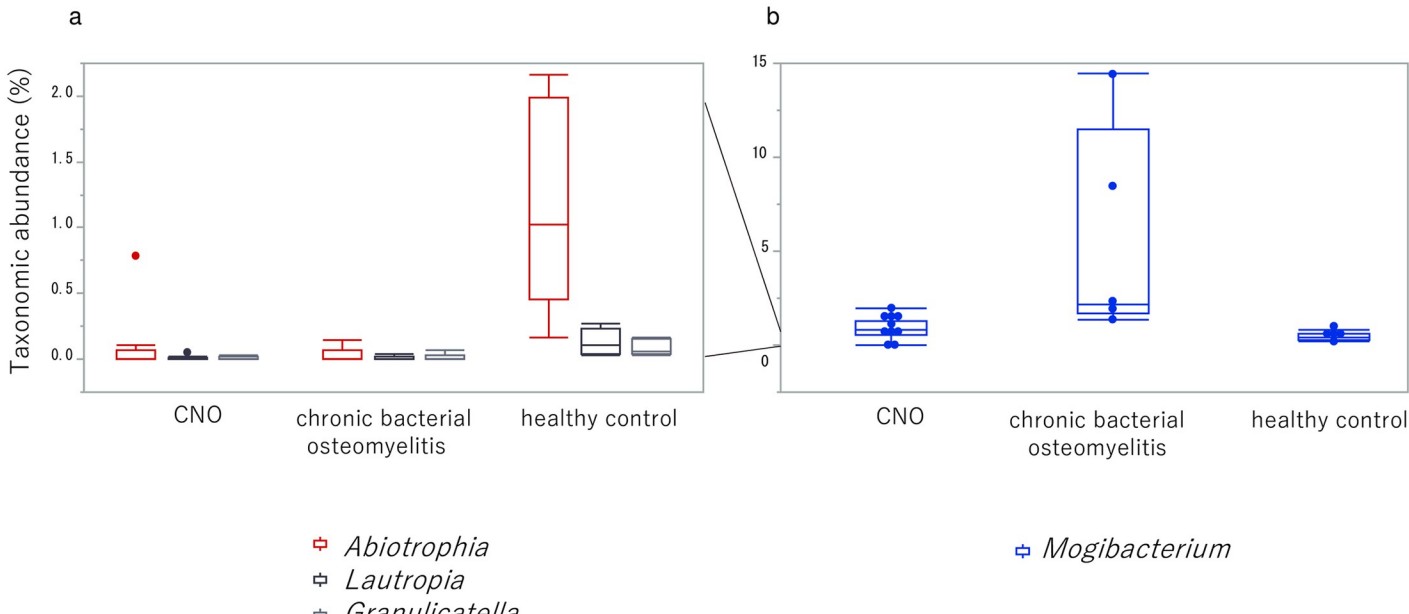

**Fig 1. Box plots of genera showing statistically significant difference in taxonomic abundance.** (a) Genera with a significant decrease in the CNO and chronic bacterial osteomyelitis groups. The Y-axis is scaled from 0 to 2% (compared to 0 to 15% of (b)) to accommodate the low relative abundance of these genera. (b) *Mogibacterium* taxonomic abundance with a significant increase in the chronic bacterial osteomyelitis group. In the box plot, the bottom and top of the box indicate 25th and 75th percentile, respectively, and the horizontal line in the box indicates median. The outliers in (a) are depicted as dots, which are located above the 75th percentile + 1.5 × interquartile range. Data for each individual in (b) are represented as dots.

*Mogibacterium* were 0.9%, 5.7%, and 0.4% in the CNO, chronic bacterial osteomyelitis, and healthy control groups, respectively. These values were calculated from individual taxonomic abundances, as illustrated in Fig 1 separately for each group. The taxonomic abundance of *Mogibacterium* was zero only in one of the CNO samples (S2 Table).

Overall, among the 74 bacterial genera, the taxonomic profiles were highly correlated between the CNO and chronic bacterial osteomyelitis groups, healthy control and CNO groups, and healthy control and chronic bacterial osteomyelitis groups (Spearman's correlation coefficient: 0.85, 0.72, and 0.77, respectively). Functional profiling did not reveal any gene family with a statistically significant increase (at a significance level of $P_{FDR}$ 0.05) in either the CNO or the chronic bacterial osteomyelitis group. In contrast, four gene families exhibited a significant decrease in the CNO group: A0A0K2RYG8 (uncharacterized protein) in *Rothia*, and A0A133S3U6 (uncharacterized protein), A0A150NXH2 (l-proline glycine betaine-binding ABC transporter protein ProX/osmotic adaptation), and D3H673 (bacteriocin immunity protein) in *Streptococcus*. These gene families were also decreased in the chronic bacterial osteomyelitis group, although to a lesser extent than in the CNO group, with $P_{FDR}$ values ranging from 0.06 to 0.1.

## Comparison between taxonomic and sequence abundance estimated by taxonomic profiling

Fig 2 depicts the overall phylogenetic profile in terms of both taxonomic and sequence abundances within the CNO, chronic bacterial osteomyelitis, and healthy controls groups. Importance of distinction between the two types of relative abundance has rarely been considered in previous studies, and was recently pointed out [29]. It showcases 12 genera and other taxa and

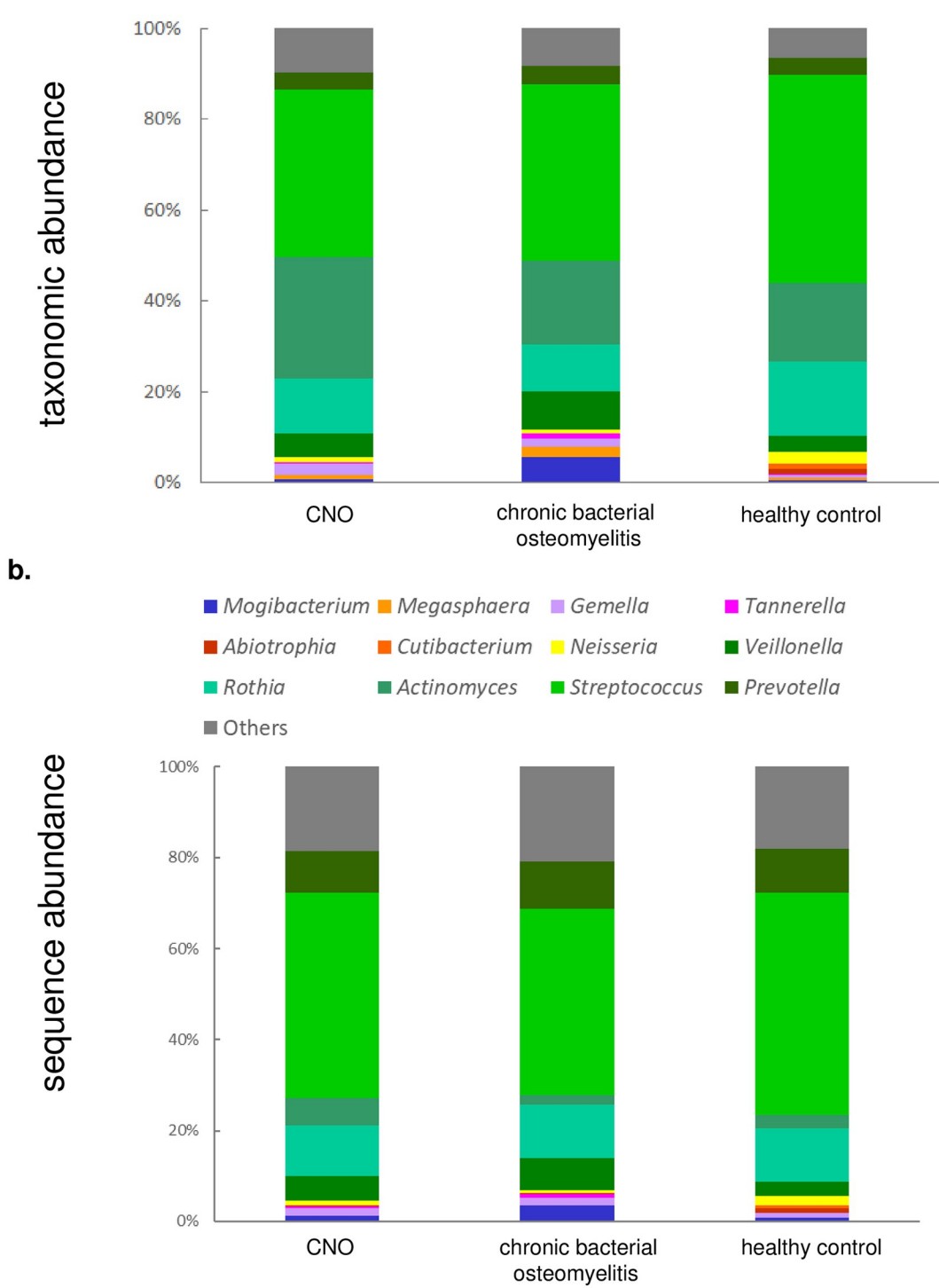

**Fig 2. Relative abundance of microbes in CNO, chronic bacterial osteomyelitis, and healthy control samples.** (a) Taxonomic abundance and (b) sequence abundance. The average values are presented for the 12 selected genera, while the remaining genera are grouped as "Others" (gray). The 12 genera displayed at least 3% taxonomic abundance in the healthy control group or more than a 1% increase or decrease on average in the chronic bacterial osteomyelitis group compared to the healthy control group. *Mogibacterium* is colored in blue. The three most abundant genera with >10% taxonomic abundance in the healthy control group (*Streptococcus*, *Actinomyces*, *and Rothia*) are denoted by greenish colors.

highlights the top 12 with the largest absolute differences in taxonomic abundance between chronic bacterial osteomyelitis and healthy controls. The result confirmed that *Mogibacterium* was consistently enriched in the chronic bacterial osteomyelitis group, even when sequence abundance was used instead of taxonomic abundance (Fig 2b).

The relationship between taxonomic and sequence abundance among the 74 genera detected in each sample is plotted in S1 Fig (raw data are in S3 Table). The Spearman's correlation coefficient for this comparison was 0.5. Several genera exhibited notable differences between taxonomic and sequence abundance estimates. *Streptococcus* and *Prevotella* displayed an overestimation of sequence abundance (>10%) compared with taxonomic abundance, whereas *Actinomyces* and *Lachnospiraceae* exhibited an underestimation of sequence abundance (>10%). These phenomena were consistently observed across multiple samples for *Streptococcus*, *Prevotella*, and *Actinomyces*. Notably, 60% of the 20 samples exhibited a statistically significant enrichment of *Actionomyces*, with a >10% underestimation in sequence abundance compared to other genera ($P<10^{-15}$, Fisher's exact test). *Streptococcus* and *Prevotella* also exhibited a statistically significant enrichment of >10% overestimation of sequence abundance ($P<10^{-6}$, Fisher's exact test).

However, *Rothia* demonstrated a contrasting pattern, with one sample showing a >10% overestimation of sequence abundance and two samples showing a >10% underestimation. The disparity in direction across these three samples may be attributed to the considerable variation in the proportion of reads that were not mapped to any taxon during the calculation of sequence abundance. Specifically, in the sample showing an overestimation, the proportion of unmapped reads was 38.0%, whereas in the two samples showing an underestimation, the proportions were 27.0% and 8.2%.

Overall, these results were consistent across the CNO, chronic bacterial osteomyelitis, and healthy control groups, thereby providing a technical basis for interpreting sequence and taxonomic abundance in the oral microbiome.

## Comparison with a previously reported core saliva microbiome genera

A previous meta-analysis conducted in 2021 [42] identified 59 core genera consistently observed in salivary shotgun metagenome sequencing projects (deposited in the MG-RAST database) and the Human Oral Microbiome Database (gray in Fig 3), in which 20 core genera were also consistently observed by amplicon sequencing (green in Fig 3). The presence of each of the 74 genera we detected (blue in Fig 3) in the previously reported core genera is shown in S2 Table. Among the previously reported 59 core genera, 32 (i.e., 9 + 23 in Fig 3) were not detected in the present salivary shotgun metagenome study, including *Acinetobacter*, *Bacillus*, *Delftia*, *Enterobacter*, *Moraxella*, *Mycobacterium*, *Pseudomonas*, *Ralstonia*, and *Sphingomonas*.

However, of the 74 genera detected in our study, 47 (including *Mogibacterium*, shown on the left in Fig 3) were not included in the 59 genera reported in the previous meta-analysis [42]. Additionally, the previous meta-analysis detected 235 core genera in the salivary shotgun metagenome sequencing projects alone, and *Mogibacterium* was not included on this list. These findings suggest that *Mogibacterium* is not a core genus in the saliva microbiome but appears to increase in abundance in chronic bacterial osteomyelitis. In addition, because the previous meta-analysis was confined to salivary shotgun metagenome data obtained in California, the substantial difference in taxon composition may reflect differences in dietary culture between North America and Japan because diet plays a substantial role in the formation of the general oral microbiome.

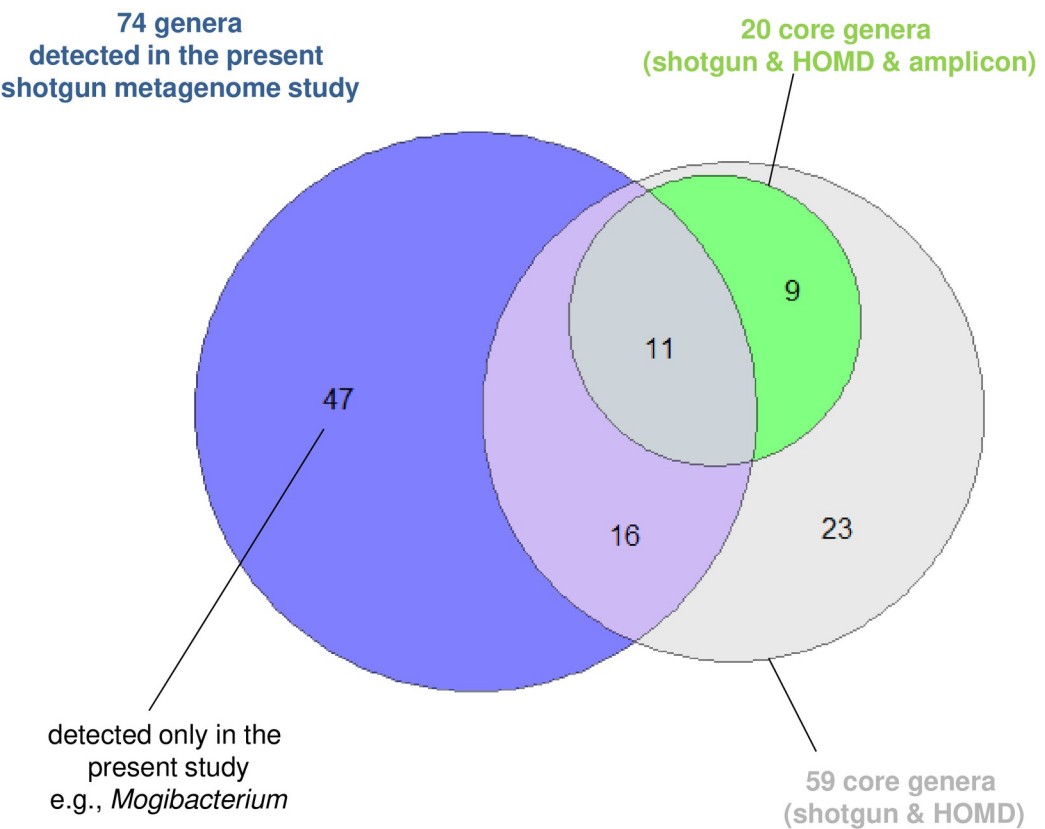

**Fig 3. Venn diagram illustrating the relationship between genera detected in the present and previous studies.**
HOMD: Human Oral Microbiome Database.

## *Mogibacterium* from chronic bacterial osteomyelitis and healthy control samples exhibits no discernible genomic differences

Assembled contigs taxonomically assigned to *Mogibacterium* were obtained in all chronic bacterial osteomyelitis samples, with total lengths ranging approximately from 1.6 to 3.5 Mbp. These contigs may include multiple *Mogibacterium* incomplete genomes per sample. Based on the gene finding of the contigs and those obtained in healthy control samples and a publicly available reference complete genome of *Mogibacterium diversum* CCUG 47132, we conducted a pan-genome analysis to create a gene presence or absence matrix. Subsequently, a maximum-likelihood core-genome tree of *Mogibacterium* (Fig 4) was constructed from the alignment of concatenated core genes (i.e., core-genome alignment). The tree indicates that the samples from chronic bacterial osteomyelitis (blue) do not form a distinct cluster compared to the healthy control samples. The core genome contained 10853 base pairs (0.6% of the reference complete genome) and 2712 single nucleotide polymorphisms, suggesting incomplete assembly of the metagenome samples and large genetic diversity in the genus. The inclusion of 10 additional CNO samples reduced the number of core genes to zero, making it impossible to construct a core-genome alignment. In addition, the pan-genome analysis revealed no unique genes specific to chronic bacterial osteomyelitis samples, confirming the genomic similarity between the two groups. These results indicated that *Mogibacterium* from chronic bacterial

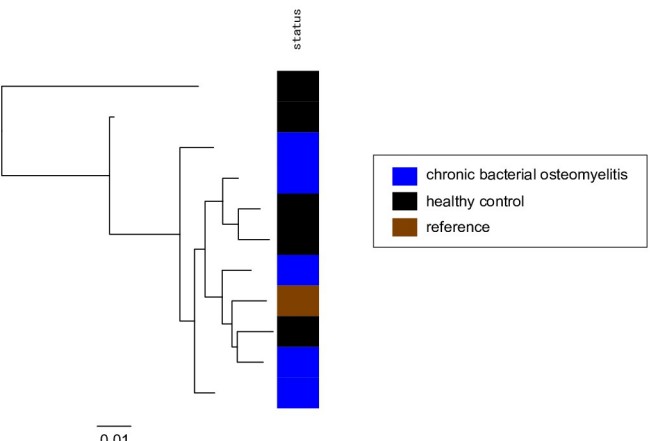

**Fig 4. Core-genome tree of *Mogibacterium*.** Nucleotide sequences from chronic bacterial osteomyelitis (blue) and healthy control (black) metagenome samples as well as the reference genome (brown) were analyzed. Sequences from the metagenome samples were obtained from assembled contigs. The scale bar represents the number of substitutions per core genome.

osteomyelitis and healthy control samples were indistinguishable at both the core- and pan-genome levels.

## Discussion

*Mogibacterium* is a genus of anaerobic, gram-positive, non-spore-forming, rod-shaped bacteria originally isolated from the periodontal pockets of adult human patients with periodontal disease and infected root canals [43]. The pro-inflammatory effects of *Mogibacterium*, *Porphyromonas*, and *Treponema* has been suggested to be responsible for the progression of medication-related osteonecrosis of the jaw, which has similar symptoms to osteomyelitis [44]. *Mogibacterium*, *Solobacterium*, *Slackia*, and *Moryella* are increased in the salivary microbiome of patients with immunoglobulin G4-related diseases compared with that of patients with primary Sjögren's syndrome [45]. Therefore, *Mogibacterium* is likely associated with multiple diseases, including chronic bacterial osteomyelitis, as shown in this study. We found an increased abundance of *Mogibacterium* in chronic bacterial osteomyelitis, suggesting its association with the disease. Further investigation of its molecular mechanisms requires experimental analysis using isolated strains, which is challenging owing to the low abundance and anaerobic culturing requirements of *Mogibacterium*.

Interestingly, despite being considered distinct diseases, CNO and chronic bacterial osteomyelitis exhibited a high overall taxonomic profile correlation (Spearman's correlation coefficient: 0.85). The only exception was for *Mogibacterium*, which showed a significant increase in the chronic bacterial osteomyelitis group, making it a potential marker for distinguishing between the two conditions. This might be of particular importance at an early stage, in which the differences between both diseases are unclear. Alternatively, differences in host immunity and/or genetics may play a more significant role in differentiating CNO and chronic bacterial osteomyelitis.

However, the sample size of the present study was small (N = 5 in the chronic bacterial osteomyelitis group). Although all five patients with chronic bacterial osteomyelitis showed a greater taxonomic abundance of *Mogibacterium* than the five healthy controls, the abundance values for two of these patients appeared to be outliers that were distinct relative to those of

CNO patients (Fig 1b). Further studies should be performed with additional data on chronic bacterial osteomyelitis patients to explore the external validity of the study and determine the frequency of such outliers, which may represent a potential characteristic of chronic bacterial osteomyelitis.

We compared the taxonomic and sequence abundances of the 74 bacterial genera detected using salivary shotgun metagenome data. Viruses were not analyzed because the summation of their sequence abundance per sample was consistently less than 0.1% and MetaPhlAn3 was not designed to estimate the taxonomic abundance of viruses. The challenges associated with sequence abundance, including its dependence on genome size and the potential for under- or overestimation compared to taxonomic abundance, have recently been highlighted using simulated data [29]. We observed that *Streptococcus* and *Prevotella* were overestimated in multiple samples and *Actinomyces* was consistently underestimated. However, the genome size of *Actinomyces* is approximately 3 Mb, which is larger than that of *Streptococcus* and *Prevotella* (approximately 2 Mb). This finding is inconsistent with that of a recent study [29], which suggested the possibility of overestimating sequence abundance in taxa with larger genome sizes. Moreover, this discrepancy may be attributed to the sequence abundance being estimated at the genus level in the current study, rather than the species level as performed in the recent study. Alternatively, this discrepancy may also be attributed to other unknown factors.

A comparison with a previously reported core salivary microbiome revealed discrepancies, with nearly half of the previously reported genera not observed in our study and more than half of our detected genera not included in the previous study. This highlights the greater taxonomic diversity of the human oral microbiota than previously recognized, which is based on the influence of factors such as geography, diet, host health conditions, and sequencing protocol.

## Conclusions

This study is the first to demonstrate a significant increase in the abundance of *Mogibacterium* in chronic bacterial osteomyelitis. Additionally, we present a systematic comparison of the taxonomic and sequence abundances of all genera detected using deep salivary shotgun metagenome data. Because the sample size used in this exploratory and descriptive study was small, these preliminary data require further verification. However, the findings of our study lay the groundwork for further investigations of the mechanisms underlying chronic bacterial osteomyelitis compared with those underlying CNO. Overall, this research contributes to a better understanding of the role of *Mogibacterium* in chronic bacterial osteomyelitis and provides a foundation for future studies aimed at unraveling the complexities of oral microbiota-related diseases.

## Supporting information

**S1 Fig. Comparison between the taxonomic and sequence abundance.** Genera showing a difference of more than 10% are highlighted in this plot.
(PPTX)

**S1 Table. List of individuals included in this study along with information regarding the metagenome sequence data.**
(XLSX)

**S2 Table. Taxonomic abundance of each genus detected in the present study.**
(XLSX)

**S3 Table. Source data for S1 Table.**
(XLSX)

# Acknowledgments

We would like to thank the Human Genome Center of the Institute of Medical Science (University of Tokyo) and at the National Institute of Genetics, where computational calculations were performed.

# Author Contributions

**Conceptualization:** Hiroko Yahara, Koji Yahara.

**Data curation:** Hiroko Yahara, Souichi Yanamoto, Miho Takahashi, Yuji Hamada, Takuya Asaka, Yoshimasa Kitagawa, Kuniyasu Moridera, Kazuma Noguchi, Yutaka Maruoka, Koji Yahara.

**Formal analysis:** Koji Yahara.

**Funding acquisition:** Hiroko Yahara.

**Investigation:** Hiroko Yahara, Koji Yahara.

**Methodology:** Hiroko Yahara, Koji Yahara.

**Project administration:** Hiroko Yahara, Koji Yahara.

**Resources:** Koji Yahara.

**Software:** Koji Yahara.

**Supervision:** Hiroko Yahara, Koji Yahara.

**Validation:** Koji Yahara.

**Visualization:** Koji Yahara.

**Writing – original draft:** Hiroko Yahara, Koji Yahara.

**Writing – review & editing:** Koji Yahara.

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
