## [Decision Letter · Decision Letter 0]

19 Feb 2024

PONE-D-23-38526Shotgun metagenomics comparing taxonomic and sequence abundances identifies Mogibacterium associated with chronic bacterial osteomyelitis in the saliva microbiomePLOS ONE

Dear Dr. Yahara,

Thank you for submitting your manuscript to PLOS ONE. After careful consideration, we feel that it has merit but does not fully meet PLOS ONE’s publication criteria as it currently stands. Therefore, we invite you to submit a revised version of the manuscript that addresses the points raised during the review process.

**ACADEMIC EDITOR:** Dear Dr Yahara,

We appreciate you submitting your manuscript to PLOS ONE and thank you for giving us the opportunity to consider your work.

I have completed my evaluation of your manuscript, which has been reviewed by two highly qualified reviewers all of whom agree it is worth to be published in PLOS ONE. Nevertheless, they have suggested some changes that will help to improve the paper.

Therefore, I invite you to resubmit your manuscript after addressing the reviewers’ comments below. When revising your manuscript, please consider all issues mentioned in the reviewers' comments carefully: please, outline every change made in response to their comments and provide suitable rebuttals for any comments not addressed. Please, note that your revised submission may need to be re-reviewed.

PLOS ONE values your contribution and I look forward to receiving your revised manuscript.

Yours sincerely,

Dr. Olga Spekker==============================

We look forward to receiving your revised manuscript.

Kind regards,

Olga Spekker, Ph.D.

Academic Editor

PLOS ONE

A clean copy of the edited manuscript (uploaded as the new *manuscript* file)”.

 [This study was supported by a Grant-in-Aid for Scientific Research of Education, Culture, Science, Sports, and Technology (MEXT) from Japan (19J40070 to H.Y.).  This work was supported in part by Grants-in-Aid for Research from the National Center for Global Health and Medicine (23A3001)].  

Reviewers' comments:

Reviewer's Responses to Questions

**Comments to the Author**

1. Is the manuscript technically sound, and do the data support the conclusions?

Reviewer #1: Yes

Reviewer #2: Yes

2. Has the statistical analysis been performed appropriately and rigorously? 

Reviewer #1: Yes

Reviewer #2: I Don't Know

3. Have the authors made all data underlying the findings in their manuscript fully available?

Reviewer #1: Yes

Reviewer #2: Yes

4. Is the manuscript presented in an intelligible fashion and written in standard English?

Reviewer #1: Yes

Reviewer #2: Yes

5. Review Comments to the Author

Reviewer #1: The authors have presented a well-written, clear and concise paper to show the results of a study to investigate the salivary bacterial composition of people with osteomyelitis. The abstract and introduction are logical and complete, the materials and methods present the sampling and data creation well, and the results and discussion bring the work together and explain the conclusions well. My only comment is the number of samples that were taken, being n=10 and n=5 respectively for different types of osteomyelitis. It is a small sample set, but I do understand that this is an exploratory and descriptive study. Are there other datasets to compare these results with? I was pleased to see the depth of sequencing- in my experience the hardest consequence of salivary studies is to gain enough bacterial reads above the human reads. A final question is what the overall bacterial abundance is in real terms and how this relates to the healthy controls. Could increase in relative abundance of Mogibacterium be a consequence of a proliferating bacterial counts in the saliva? In short, a complete descriptive paper that will likely stimulate understandings of the progression of disease.

Reviewer #2: Review of the article titled Shotgun metagenomics comparing taxonomic and sequence abundances identifies Mogibacterium associated with chronic bacterial osteomyelitis in the saliva microbiome

Summary

The authors of this article clearly display the signs of expertise and competence in their field. The article is well written and the main points are clearly discussed with understandable and elaborate language. Based on their published literature they are well entrenched in this research field, which is clearly shown from their confidence in handling their data and discussing the results. I believe that the findings of this article are appropriate for publication and they will be a valuable addition in the oral microbiome research. I would like them however to address some of my concerns first, mainly about the statistical analysis that they employed.

Major revisions

My biggest problem with the article is the data analysis in the Taxonomic profiling identifies microbes associated with chronic bacterial osteomyelitis section. Seemingly the “taxonomic abundance” data that you publish in this article (Table S2) is not exactly the same as the data that were used in your statistical analyses that you discuss. I take this from your statement in Line 228-229 in parenthesis and from the fact that I tried to replicate your analyses using Table S2 and I obtained substantially different results.

I performed pairwise Wilcoxon tests between the patient groups across all taxa which yielded more than 4 significant differences with the given 0.05 significance threshold or none with a Bonferroni corrected threshold. I also tried to replicate your boxplots from Figure 1 but I found interesting differences (the position of the median line, position of your data points between the 25th and 75th percentiles), which – for me – suggest that there was an underlying distribution (differing from the data published in Table S2) that was used to create these boxplots.

The reason for these discrepancies are not clear to me. If there is a difference between the employed data in the analyses and the published data in Table S2 then this difference should be strongly emphasized in the text. I would also like you to include a more detailed description of the data handling steps mentioned in Line 229 (“using a model with default normalization and transformation implemented in MaAsLin2”) in the Methods section, as this would explain why you would get different results from the raw sequence data.

If, however, I should be able to replicate your results from the available data table (Table S2) alone, then please explain to me why I obtained different results from the ones you discussed in this section. For clarity I include my R script (verification.R) for you to check the validity of my claims.

I had another problem specifically with the results concerning the Mogibacterium comparison. In my R script I also try to show you that assuming a simplistic normal distribution, 2 of your samples seems highly likely to be outliers. I don’t want to assume that this type of data is necessarily normally distributed, but this should still indicate that you should be careful interpreting these results. I see that you touch on this topic in Line 407-408, but I would like you to also include a short disclaimer in this section as well, that you are indeed aware of the low sample size of the chronic bacterial osteomyelitis group and the possibility of some samples to be outliers.

Minor revisions

Title – I would suggest a modified title “Shotgun metagenomic analysis of saliva microbiome suggests Mogibacterium as a factor associated with chronic bacterial osteomyelitis” or something along this line. As I don’t think your discussion of taxonomic and sequence abundance comparison is that central to this articles message. I also suggested to change the “identify” part as you yourself describe these results as preliminary in the Conclusion section (Line 407-408).

Line 39 “previous study” – Please include a reference here.

Line 168-170 – “we randomly selected the same number of read pairs [..] using BBmap.” – If this is the standard method when account for differences in sequence coverage than please include a reference here where they discuss the reason behind this method. If this is a new approach than please include a description here about the validity of this method. Also I think a reference is missing here for BBmap.

Line 176-177 Which program was used for the concatenation?

Line 186-187 “Contig Annotation Tool (CAT) [30]” – I think a reference is missing from here, because the Nakazawa et al. 2002 article is referenced here which clearly belong to Line 196 where it is given appropriately. Please insert the proper reference (this would be von Meijenfeldt et al. 2019, https://doi.org/10.1186/s13059-019-1817-x ?) and number the rest of the references accordingly.

Line 194-195 Mogibacterium is not mentioned prior to this statement, maybe a short introduction should be included to justify why this specific taxon was selected for the phylogenetic analysis.

Line 202-203 Where can I find this heatmap, please reference this statement. If the picture/table is not included in the article, why do you mention it?

Line 209-210 I think it would be appropriate to reference both JMP Pro and R software as well in the Reference list to clarify the availability of these softwares.

Line 253-256 “were highly correlated (Spearman’s correlation coefficient: 0.85) […]” – the other groups seemed also highly correlated, I am not sure whether this difference is high enough to justify discussion

Line 284-285 “The blue line indicates the significant enrichment […]” – I cannot see a blue line on Figure 2. Please clarify the meaning of the sentence.

Line 393-395 “[…] using multiple reference genomes across species, […]” – This approach is first mentioned here. Please include a more detailed description of the validity of this method compared to others in the Method section.

TableS2 The heading of the first column describes sampleIDs, the table however seemingly contains genus names. If it was the intended nomenclature, please clarify why you chose “sampleID”, otherwise correct the header.

Spelling errors

Line 43 “level” – levels

Line 44 “in a chronic” – in the chronic

Line 45-47 “this is the first study…” – this seems like a strong statement, maybe it would be prudent to lighten it

Line 81 “chronic specific inflammation” – I don’t think chronic specific is an appropriate term in English. Maybe “CNO is characterized by markers specific to chronic inflammation,” or something along this line.

Line 96 “For instance, a” – an

Line 122 “from previous out study” – from our previous study

Line 133 “techniques have employed” – are employed

Line 148-149 “A total of 15 patients […] with CNO in Japan”. – “A total of 15 Japanese patients were […]”; or “A total of 15 patients were chosen from Japan for this study […]”. The original statement reads like only the 10 CNO patients are from Japan.

Line 160 “data of the five healthy control” – data of five healthy controls

Line 167 “Afte” – After

Line 338 “which may may include” – which may include

Questions

Line 327-328 – I am not sure about the origin of the data used in the Oliveira et al. 2021 article, but I would presume it analysed the salival microbiomes of mainly North-American and Western European individuals. Can the difference in the taxon composition of your data also be the result of the different dietary culture of Japan and/or East Asia? I would assume that diet plays a substantial role in the formation of the general oral microbiome. I see that you elaborate on this a bit in Line 400-401, but I think a short paragraph could be also included about this after Line 328. Actually, your finding of these completely different genera compared to the previously reported “core salival microbiome” is much more interesting than your discussion of the Mogibacterium related results. Maybe you could elaborate on these findings in a subsequent article.

6. PLOS authors have the option to publish the peer review history of their article (what does this mean?). If published, this will include your full peer review and any attached files.

Reviewer #1: **Yes: **Kate Howell

Reviewer #2: No

---

## [Author Response · Author response to Decision Letter 0]

4 Mar 2024

Uploaded as a Word file and included at the end of the combined PDF

---

## [Decision Letter · Decision Letter 1]

26 Mar 2024

PONE-D-23-38526R1Shotgun metagenomic analysis of the saliva microbiome suggests Mogibacterium as a factor associated with chronic bacterial osteomyelitisPLOS ONE

Dear Dr. Yahara,

Thank you for submitting your manuscript to PLOS ONE. After careful consideration, we feel that it has merit but does not fully meet PLOS ONE’s publication criteria as it currently stands. Therefore, we invite you to submit a revised version of the manuscript that addresses the points raised during the review process.

We look forward to receiving your revised manuscript.

Kind regards,

Olga Spekker, Ph.D.

Academic Editor

PLOS ONE

Journal Requirements:

**Additional Editor Comments:**==============================

**ACADEMIC EDITOR:**

Dear Dr. Yahara,

We appreciate you submitting your manuscript to PLOS ONE and thank you for giving us the opportunity to consider your work.

I have completed my evaluation of your manuscript, which has been reviewed by two highly qualified reviewers all of whom agree it is worth to be published in PLOS ONE. Nevertheless, one of the reviewers has suggested some minor changes regarding the supplementary tables.

Therefore, I invite you to resubmit your manuscript after addressing these minor comments below.

PLOS ONE values your contribution and I look forward to receiving your revised manuscript.

Yours sincerely,

Dr. Olga Spekker

Reviewers' comments:

Reviewer's Responses to Questions

**Comments to the Author**

1. If the authors have adequately addressed your comments raised in a previous round of review and you feel that this manuscript is now acceptable for publication, you may indicate that here to bypass the “Comments to the Author” section, enter your conflict of interest statement in the “Confidential to Editor” section, and submit your "Accept" recommendation.

Reviewer #1: All comments have been addressed

Reviewer #2: All comments have been addressed

2. Is the manuscript technically sound, and do the data support the conclusions?

Reviewer #1: (No Response)

Reviewer #2: Yes

3. Has the statistical analysis been performed appropriately and rigorously? 

Reviewer #1: (No Response)

Reviewer #2: Yes

4. Have the authors made all data underlying the findings in their manuscript fully available?

Reviewer #1: (No Response)

Reviewer #2: Yes

5. Is the manuscript presented in an intelligible fashion and written in standard English?

Reviewer #1: (No Response)

Reviewer #2: Yes

6. Review Comments to the Author

Reviewer #1: (No Response)

Reviewer #2: Answer to revisions

Summary

Thank you for considering my review suggestion. I think the presentation of your data improved significantly and the inclusion of the requested paragraph clarified the background workings of your analysis framework. The language and overall structure of your article is still very clear and straightforward.

In the next section, I would like to answer the answers to my reviews. Then I listed some suggestions concerning the Supplementary Tables. I wouldn’t like an answer for these I just included them as to make sure to really bring out the maximum from your excellent article.

Answers

"The comparison also shows that the data points are the same between the 25th and 75th percentiles (shown as black dots in the middle boxes). Although the 75th percentile is indeed different (11.4 in Fig 1b and 8.5 in the figure created by reviewer’s R script), we confirmed that it is due to a difference in the quantile algorithm between JMP and R: the 75th percentile in Fig 1b (11.4) calculated by JMP is obtained using the 6th type quantile algorithm implemented in R, whereas that in the figure by the reviewer’s R script (8.5) is obtained using the 7th type quantile algorithm, which is the default in R."

Thank you for this thorough explanation I didn’t even know that quartile calculation was this complicated in R. I accept your answer.

"According to your suggestion, we have added the following to the Methods: “According to previous studies [38, 39], we specified the total sum scaling (TSS) normalization (to ensure that the profile values ranged from 0 to 1) and arcsine square-root transformation (AST) (to stabilize the variance and improve parametric estimation models in the presence of violated data assumptions, such as normality and homoscedasticity) [37].” (Lines 216–219) We have also confirmed that the arcsine square-root transformation (AST) is conducted using the following R code "sign(x) * asin(sqrt(abs(x)))", and the taxonomic abundance of Mogibacterium (normalized to be from 0 to 1) before and after the transformation is distributed as shown below (left and right, respectively). The transformation seems to make the distribution less skewed."

Thank your for including this section in Methods. Your explanation clarifies the exact method that you were using, and thus I accept your answer. This helps to facilitate reproducibility furthermore provides opportunities for you to show your seemingly considerable experience working with this kind of data, which – I think – overall strengthens the viewers assessment of your ability to produce trustful results.

"We appreciate your comment and agree with it. The illustration of the distribution above shows that the two samples at the right seem to be outliers, even after the arcsine square-root transformation (AST). Therefore, we added the following to the Discussion: “However, the sample size of the present study was small (N=5 in the chronic bacterial osteomyelitis group). Although all five patients with chronic bacterial osteomyelitis showed a greater taxonomic abundance of Mogibacterium than the five healthy controls, the abundance values for two of these patients appeared to be outliers that were only distinct relative to those of CNO patients (Fig 1b). Further studies should be performed with additional data on chronic bacterial osteomyelitis patients to explore the external validity of the study and determine the frequency of such outliers, which may represent a potential characteristic of chronic bacterial osteomyelitis. (Lines 390–397)"

Thank you again, this section further strengthens the sincerity and transparency of your previous statements.

All together I think this article is appropriate for publication in this state.

Suggestions

The first row in Table_S1 contains 3 white spaces before the beginning of the sentence.

The name of the 5th control individual is CN001_ with an ending underscore is this a typo?

The genus header in Table_S2 is still sampleID in the uploaded file but I presume you just didn’t upload the revised version along the Response to reviewers file.

I am sure the publisher has regulations in this regard, however I saw some really bad examples in even highly regarded publishers like Nature, that the name of the supplementary files were jumbled completely. Please name your files in a straightforward manner like Table_S1 or something like that because this really helps later reading and organisation for your readership.

7. PLOS authors have the option to publish the peer review history of their article (what does this mean?). If published, this will include your full peer review and any attached files.

Reviewer #1: No

Reviewer #2: No

---

## [Editor Report · Decision Letter 2]

9 Apr 2024

Shotgun metagenomic analysis of the saliva microbiome suggests Mogibacterium as a factor associated with chronic bacterial osteomyelitis

PONE-D-23-38526R2

Dear Dr. Yahara,

We’re pleased to inform you that your manuscript has been judged scientifically suitable for publication and will be formally accepted for publication once it meets all outstanding technical requirements.

Kind regards,

Olga Spekker, Ph.D.

Academic Editor

PLOS ONE